# Large Benefit from Simple Things: High-Dose Vitamin A Improves *RBP4*-Related Retinal Dystrophy

**DOI:** 10.3390/ijms23126590

**Published:** 2022-06-13

**Authors:** Vasily M. Smirnov, Baptiste Wilmet, Marco Nassisi, Christel Condroyer, Aline Antonio, Camille Andrieu, Céline Devisme, Serge Sancho, José-Alain Sahel, Christina Zeitz, Isabelle Audo

**Affiliations:** 1Sorbonne Université, INSERM, CNRS, Institut de la Vision, F-75012 Paris, France; vasily.smirnov@inserm.fr (V.M.S.); baptiste.wilmet@inserm.fr (B.W.); marco.nassisi@inserm.fr (M.N.); christel.condroyer@inserm.fr (C.C.); aline.antonio@inserm.fr (A.A.); j.sahel@gmail.com (J.-A.S.); 2Faculté de Médecine, Université de Lille, F-59000 Lille, France; 3Exploration de la Vision et Neuro-Ophtalmologie, CHU de Lille, F-59000 Lille, France; 4Centre Hospitalier National d’Ophtalmologie des Quinze-Vingts, National Rare Disease Center REFERET and INSERM-DGOS CIC 1423, F-75012 Paris, France; candrieu@15-20.fr (C.A.); cdevisme@15-20.fr (C.D.); ssancho@15-20.fr (S.S.); 5Department of Ophthalmology, University of Pittsburgh School of Medicine, Pittsburg, PA 15213, USA

**Keywords:** inherited retinal degeneration, fundus albipunctatus, retinol-binding protein, RBP4, retinol treatment

## Abstract

Inherited retinal diseases (IRD) are a group of heterogeneous disorders, most of which lead to blindness with limited therapeutic options. Pathogenic variants in *RBP4*, coding for a major blood carrier of retinol, retinol-binding protein 4, are responsible for a peculiar form of IRD. The aim of this study was to investigate if retinal function of an *RBP4*-related IRD patient can be improved by retinol administration. Our patient presented a peculiar white-dot retinopathy, reminiscent of vitamin A deficient retinopathy. Using a customized next generation sequencing (NGS) IRD panel we discovered a novel loss-of-function homozygous pathogenic variant in *RBP4*: c.255G >A, p.(Trp85*). Western blotting revealed the absence of RBP4 protein in the patient’s serum. Blood retinol levels were undetectable. The patient was put on a high-dose oral retinol regimen (50,000 UI twice a week). Subjective symptoms and retinal function markedly and sustainably improved at 5-months and 1-year follow-up. Here we show that this novel IRD case can be treated by oral retinol administration.

## 1. Introduction

Inherited retinal diseases are a heterogeneous group of disorders, the most common of which—retinitis pigmentosa—progresses towards blindness with limited therapeutic options. Some rare forms of IRD, linked with inborn defects of small molecule metabolism can benefit from metabolic and/or dietary treatment [1,2]. For instance, vitamin A and E intake improves vision in patients with abetalipoproteinemia (OMIM# 200100), a hereditary defect of fat-soluble vitamin absorption and transport [3]. Dietary arginine restriction and vitamin B6 intake can reduce blood ornithine level and slow retinal degeneration in patients with gyrate atrophy (OMIM#258870) [4,5]. Low phytanic acid diet may slow retinal degeneration progression rates in adult Refsum disease (OMIM#266500) [6,7].

Retinol-binding protein 4 (RBP4, Uniprot#Q5VY30) is a major blood transporter of retinol from hepatocyte to target organs. Pathogenic variants in *RBP4* are associated with both ocular developmental abnormalities and retinal degeneration (OMIM#615147). Ocular developmental abnormalities range from a mild iris coloboma to micro- and anophthalmia. Retinal degeneration consists of a rod–cone dystrophy, also known as retinitis pigmentosa.

We report here that high-dose vitamin A is able to improve visual function in a new case of *RBP4*-associated retinopathy.

## 2. Results

### 2.1. Detailed Case Description

A male patient of Algerian ancestry was initially assessed at age 12 years. He had decreased visual acuity and progressive night blindness since early childhood. Best corrected visual acuity was 20/32 for both eyes with spectacle correction −0.50(−1.0)5° in the right eye (RE) and −0.75(−0.25)170° in the left eye (LE).

Goldman kinetic visual field tested on V4e, V1e, and V1e targets was normal while the III1e and smaller targets were not perceived by the patient (Appendix A). Automated static perimetry showed a pericentral ring scotoma. Dark-adapted responses (DA0.01) of ISCEV standard full-field electroretinogram (ffERG) were undetectable while DA3.0 and DA10.0 revealed severely reduced and delayed responses; light-adapted responses (LA3.0 and LA3.0 flicker) were severely reduced with implicit time shift in keeping with generalized rod–cone dysfunction (Appendix A). Full Stimulus Threshold (FST) revealed severely reduced threshold for the white stimulus (Appendix A). Fundus examination revealed numerous white dots scattered over the mid and far periphery with no pigmentary or atrophic retinal changes (Figure 1A and Appendix A). On infrared reflectance (IRR) imaging, both maculae were granular (Figure 1B). Increased image averaging on short-wave fundus autofluorescence (SWAF) revealed an ellipsoid-shaped ring of increased autofluorescence around the fovea on an overall hypoautofluorescent background (Figure 1C). Near-infrared fundus autofluorescence (NIRAF) imaging also showed a reduced signal (Figure 1D). Spectral domain optical coherence tomography (SD-OCT) centered on the fovea revealed an hyporeflective irregular ellipsoid zone (EZ) (Figure 1E). The outer nuclear layer (ONL) thickness was preserved with an unusual hyper reflective band on both sides of the fovea (Figure 1F, yellow arrows). SD-OCT performed through the peripheral white dots revealed hyperreflective dots above the retinal pigment epithelium (RPE), interrupting the EZ (Figure 1G, white arrows).

A general examination was performed to detect any vitamin A deficiency-related alterations. The only skin issue was subtle acne vulgaris on the forehead. Past medical history was unremarkable and the patient declined any dietary restrictions.

### 2.2. Genetic and Functional Studies

The family consisted of unaffected first-degree-cousin parents of Algerian descent and two unaffected sisters (Figure 2A). Targeted NGS identified a novel homozygous nonsense variant c.255G >A, p.(Trp85*) in *RBP4* which co-segregated with disease in this family. This variant was classified as pathogenic Ia (PVS1, PP1-S, PM2) in accordance with ACMG standards [8]. Western blot analysis revealed the absence of the RBP4 protein band in the serum of the patient (Figure 2B). Blood retinol level was also undetectable.

### 2.3. Treatment and Follow-Up

After discussion with the patient and his parents and their informed consent, high-dose oral vitamin A was initiated with retinol palmitate, 50,000 UI twice a week. Liver function was monitored. Five months after initiation of the treatment, the patient reported a significant subjective night vision improvement and a better tolerance of light-to-dark transitions. There was an improvement in FST at this time point (+25% in RE and 40% in LE, respectively) sustained at 1-year follow-up (Figure 3 and Appendix A), while visual acuity, ERG, and retinal imaging (more specifically, white dots, ONL and EZ aspects) remained unchanged. We also observed a marked visual field improvement: III1e and II1e targets became perceived by the patient (Appendix A). There were no signs of general retinol toxicity and liver function remained normal.

## 3. Discussion

The visual system is highly dependent upon retinol supply. Retinoids are essential for normal ocular development as a whole and for the maintenance of a normal tissue structure. Abnormalities in nutritional supply, storage, transport, delivery, or metabolism of retinoids are linked with a broad spectrum of developmental and degenerative diseases of the eye [13].

RBP4 is the major plasma carrier of retinol and retinoids. It forms a heterohexameric complex with transthyretin in the blood, a characteristic which protects small molecules of RBP4 (21k Da) from glomerular filtration [14]. There are two different pathways for retinoids to reach their target tissues. The main pathway depends upon RBP4, which is complexed to retinol (holo-RBP4) and then recognized and bond to a membrane retinoid receptor STRA6, located at the basal side of the RPE. STRA6 allows the internalization of retinol [15]. The minor pathway is under the control of the scavenger class B type I receptor (Sr-BI) which enables the absorption of the protein-free fraction of retinoids transported within circulating lipoprotein complexes and chylomicrons. In the absence of RBP4, this second low-rate route could be sufficient to provide retinoids to all other tissues except for the RPE, leading to a retinal disease [16,17,18].

Pathogenic variants in *RBP4* are responsible for rod–cone dystrophy associated with various degrees of microphthalmia, coloboma, and comedogenic acne. Only few patients harboring RBP4 defects have been reported to date [9,10,19,20] (Figure 2C). Our patient presents a mild phenotype unlike patients harboring *RBP4* gene defects reported previously which could be explained in part by his young age (12 years) compared to the late adult cases with advanced retinal degeneration reported in the literature. The observed white dot retinopathy in our patient could be an early feature of progressive inherited retinal degeneration.

In contrast, our patient had a very unusual phenotype for RBP4 deficiency. His ocular findings were reminiscent of fundus albipunctatus (FA, OMIM#136880) due to the presence of night blindness, numerous peripheral white dots, and no signs of retinal degeneration. FA clinical phenotype is typically associated with pathogenic variants in *RDH5* (OMIM# 601617). However, retinal white dots in our patient were less numerous and clearly not organized in a network pattern as in *RDH5*-retinopathy. The ERG was also different showing a generalized severe rod–cone dysfunction distinct from the Riggs-type ERG [21] classically associated with *RDH5*-retinopathy including normal cone function and cone-dominated dark-adapted responses. Retinol-dehydrogenase 5 is a visual cycle enzyme oxidizing 11-cis-retinol into 11-cis-retinal. The lack of enzyme activity leads to 11-cis and 13-cis retinyl esters accumulation which are thought to be the origin of the white dots in FA [22,23].

Another retinal disease very close to our patient’s presentation is retinitis punctata albescens (RPA, OMIM#136880 or Bothnia dystrophy, OMIM#607475). White dots are usually present in the early stages of the disease and are due to an accumulation of all-trans-retinyl esters in the retinal pigment epithelium secondary to an impairment of the cellular retinaldehyde–binding protein 1 (CRABP1, Uniprot#P29762), slowing down the isomerization of all-trans-retinyl esters in 11-cis-retinol. Pathogenic variants in *RLBP1*, encoding CRABP1, are responsible for this phenotype. As its name implies, RPA is a progressive retinal degeneration. Unlike in our patient, RPA is characterized by moderate narrowing of the retinal vasculature, optic disc pallor, pigmentary changes, and peripheral scalloped areas of chorioretinal atrophy [24,25]. ERG responses in RPA are, however, closer to those of our patient, with severely affected rod responses and more preserved cone responses. A limited recovery of ERG responses after prolonged dark adaptation is also reported in RPA [26] as in *RDH5*-retinopathy. This was unfortunately not tested in our patient. White dots have also been described in *RHO* (OMIM#180380) [27], *PRPH2* (OMIM#179605) [28], *LRAT* (OMIM#604863) [29,30], and *RPE65* (OMIM#180069) [31,32] gene defects, but the phenotype in these cases is more severe and progressive; thus, delineating it from our patient’s clinical picture.

Functional and morphological retinal changes in our patient were close to those reported in vitamin A deficient retinopathy (VAD). Functionally, generalized photoreceptor dysfunction with rod responses being more altered than cone responses is also characteristic for VAD [33,34]. White retinal dots can be a feature in VAD [33,34], but they are somewhat different in shape (indistinct borders) and hypoautofluorescent on SWAF [35]. Our case also presented intriguing finding with an additional hyperreflective band in the parafoveal region on OCT (Figure 2F). We hypothesize that this alteration could be a partial duplication of the outer plexiform layer and may be related to the important role of retinoids in retinal development [36].

Another inherited form of retinal degeneration linked with an altered absorption and biodistribution of vitamin A is abetalipoproteinemia (or Bassen–Kornzweig syndrome). Biallelic gene defects in *MTTP* (OMIM#157147), encoding Mitochondrial Triglyceride Transfer Protein (Uniprot#P55157) lead to impaired assembly and secretion of plasma lipoproteins that contain apolipoprotein B (very low- and low-density lipoproteins and chylomicrons). Lipoproteins facilitate absorption and carry a free fraction of fat-soluble vitamins (A, D, E, K). Upon reduced MTTP activity, there is a reduced retinol and tocopherol absorption, transport, and delivery to the target organs, including the eye, which results in retinal degeneration. Early treatment with high-dose vitamins A and E in patients with abetalipoproteinemia resulted in improvement in their retinal function and a slower retinal degeneration progression rate [3,37,38].

Oral administration of high-dose retinol palmitate has already shown to raise the level of free plasma retinol and retinyl esters in RBP4-deficient patients [39] but no visual outcome had been reported so far. Nevertheless, experimental studies report that *RBP*^-/-^ mice are able to use alternative RBP-independent pathways for retinol supply to the retina with phenotypic rescue provided by a retinol-sufficient diet [17]. In order to compensate for the lack of RBP4-related retinol transport and attempt to enhance the free fraction of retinoids delivered via the slow RBP4-independent pathway, we decided to prescribe high doses of retinol to our patient. After five months of high-dose oral vitamin A, the patient reported subjective improvement in dimly lighted environments which was supported by FST and kinetic perimetry changes from baseline. This effect further improved after one year follow-up.

Adverse and toxic effects of long-term high retinol intake are numerous including bone toxicity (hypercalcemia and osteoporosis), neurotoxicity (intracranial hypertension), and liver toxicity (liver enlargement, cirrhosis) [40]. Saturation of the RBP4-independent pathway of retinol delivery could also be detrimental for xanthophyll pigment uptake by the retina, as this slow pathway is competitive between retinol and lutein/zeaxanthin transport [41]. However, toxicity did not occur in patients with abetalipoproteinemia on prolonged high-dose retinol treatment, as the overall blood levels remained low [3]. Our patient did not experience any adverse effect after 1 year of retinol intake. Liver enzymes and blood calcium were normal. Long-term follow-up with close monitoring for vitamin A tolerability will determine whether high doses of retinol are able to prevent retinal degeneration.

## 4. Materials and Methods

### 4.1. Clinical Studies

The patient was clinically investigated at the national reference center for rare ocular diseases REFERET of the Centre Hospitalier National d’Ophtalmologie des Quinze-Vingts as previously described [42]. Briefly, best-corrected visual acuity (BCVA), refractive error, slit-lamp biomicroscopy, static and kinetic visual fields, full-field electroretinogram according to the standards of International Society for Clinical Electrophysiology of Vision [43] (Espion, Diagnosys LLC, Lowell, MA), fundus photography, spectral domain optical coherence tomography (Spectralis OCT, Heidelberg Engineering, Inc., Heidelberg, Germany), infrared and short-wavelength autofluorescence (Heidelberg Retinal Tomograph, Heidelberg Engineering, Inc.) were performed. Full stimulus threshold was assessed using achromatic full-field stimuli (Diagnosys Espion system, Diagnosys LLC, Lowell, MA, USA) as previously described [11,12].

### 4.2. Genetic Analysis

Blood samples from the index case and from his parents were collected for genetic research and genomic DNA was extracted as previously reported [44]. These DNA samples were stored and obtained from the NeuroSensCol DNA bank, for research in neuroscience (PI: JA Sahel, co-PI I Audo, partner with CHNO des Quinze-Vingts, Inserm and CNRS, certified NFS96-900). Targeted next generation sequencing (NGS) was performed in collaboration with an external company (IntegraGen, Evry, France) [45]. The *RBP4* (MIM#180250) variant selected after NGS was validated in the index case and relatives by Sanger sequencing (refseq: NM_001323517.1, primer sequences and conditions available on demand).

### 4.3. Western Blot

Western blot identification of RBP4 was performed as described in a previously published protocol [46]. Loading control was performed using anti-transferrin monoclonal antibody (Abcam, Discovery Drive, Cambridge Biomedical Campus, Cambridge, UK, ab109503).

## 5. Conclusions

We report here the first case of *RBP4*-related retinopathy manifesting as a white-dot retinopathy whose visual function improved after high-dose oral vitamin A intake. Such treatment should be considered in early stages of *RBP4*-related retinopathy. Future research will determine whether high level of vitamin A intake is able to reverse the phenotype sustainably.

## Figures and Tables

**Figure 1 ijms-23-06590-f001:**
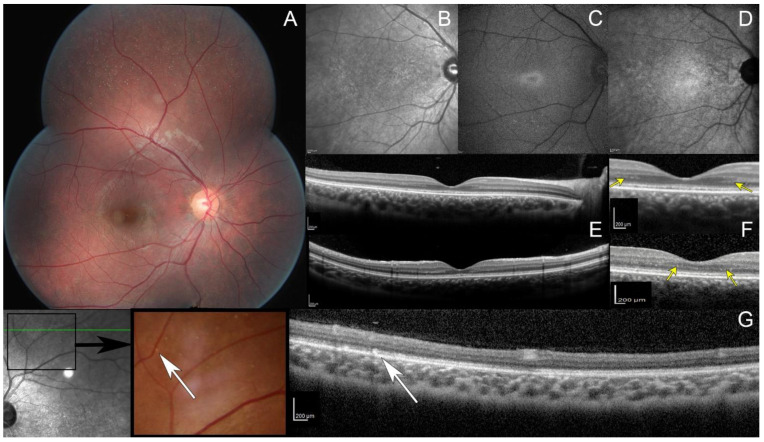
Multimodal retinal imaging. (**A**) Fundus photo, multiple white dots scattered over the midperipheral retina. Note the absence of intraretinal pigment migration and the lack of retinal vessel attenuation. (**B**) Infrared reflectance image, macular granularity. (**C**) Short-wavelength fundus autofluorescence imaging obtained with significant averaging due to the generalized reduced autofluorescence, small peri-foveal hyperautofluorescent ring with indistinct borders. (**D**) Near infrared fundus autofluorescence, small hypoautofluorescent dots. (**E**,**F**) spectral domain optic coherence tomography (SD-OCT, top: horizontal scan; and bottom: vertical scan), hypo reflective and fragmented ellipsoid zone with no interdigitation zone; preserved outer nuclear layer thickness with an unusual hyper reflective band on both sides of the fovea (yellow arrows). (**G**) OCT (scan passing through white dots, white arrows), hyperreflective dots above the retinal pigment epithelium with a focal interruption of the ellipsoid zone.

**Figure 2 ijms-23-06590-f002:**
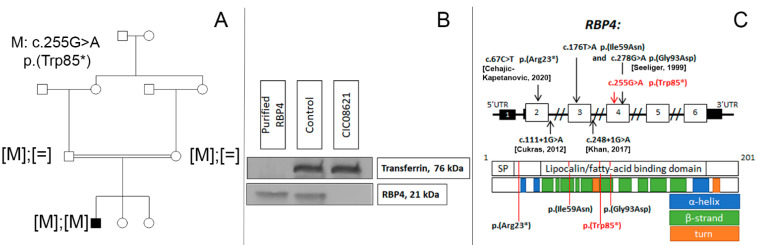
(**A**) Novel biallelic *RBP4* variant co-segregating with the disease. (**B**) Western blot analysis showing the absence of RBP4 in peripheral blood of our patient (CIC08621) carrying the homozygous nonsense variant in RBP4, compared to an unaffected control and recombinant RPB4. Transferrin is used as serum loading control. (**C**) RBP4 gene and protein structure. SP—signal peptide domain. Disulfide bonds (22-178, 88-192, 138-147) are not shown. Previously reported and novel (in red) variants linked with inherited retinal degeneration [9,10].

**Figure 3 ijms-23-06590-f003:**
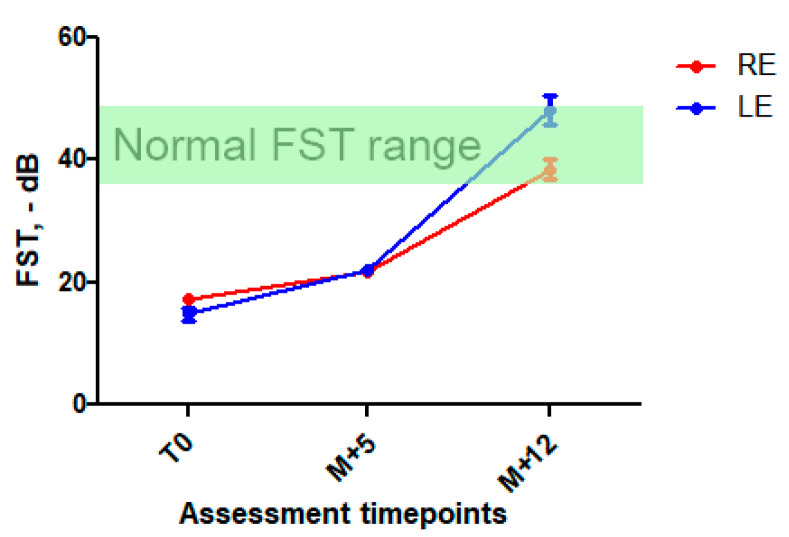
Full stimulus threshold before and after high-dose vitamin A intake. Mean FST ± SD at first assessment, at M + 5 and M + 12 follow-up. A significant improvement in FST was observed at M + 5 (25 and 40% decrease) and was sustained at M + 12 (123 et 215% in RE and LE, respectively). In green, normal FST range [11,12].

## Data Availability

All data are contained within the article or Appendix A.

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
