# Peer review of "Large Benefit from Simple Things: High-Dose Vitamin A Improves RBP4-Related Retinal Dystrophy"

_ijms, 2022, doi:10.3390/ijms23126590_

Round 1

Reviewer 2 Report

I reviewed the case report "Large benefit from simple things: high dose vitamin A improves RBP4-related retinal dystrophy". Here are my comments:

  1. The English is good, but there are several small formatting issues, like double spaces, and a major one: the discussion is reported twice.
  2. The case report does not seem groudbraking to me, as it is well documented in the literature that vitamin A correlates with benefits for retinal pathologies. To address this issue, authors should better explain the treatment scheme applied, and where the novelty of this manuscript is. 

Reviewer 3 Report

Smirnov et al. presented a case report in which high dose vitamin A administration improved one 12 years old patient with RBP-deficiency related retinal dystrophy. Overall, this manuscript was poorly conducted and presented. First of all, there is no mechanistic link indicating high dose vitamin A would benefit patient with RBP4 deficiency. Indeed, the clinical outcome is not significant.  Second, majority of the results in the manuscript lack critical experimental controls or proper statistical analysis. For example, a loading control is needed in Fig. 2A showing the absence of RBP4 in the patient. Finally, the manuscript was organized in a careless way. The language is full of typos and grammar mistakes, which will need extensive editing. The discussion section appeared twice. Therefore, this manuscript will not be suitable for publication at International Journal of Molecular Sciences.

Round 2

Reviewer 2 Report

I appreciate the explanations authors implemented, I agree with the publication of the ms.

Reviewer 3 Report

This revised version of manuscript has been improved dramatically. My concerns have been addressed and the manuscript now is suitable for publication.